# Phase Resolved Simulation of the Landau–Alber Stability Bifurcation

**Agissilaos G. Athanassoulis**

Department of Mathematics, University of Dundee, Dundee DD1 4HN, UK; aathanassoulis@dundee.ac.uk

**Abstract:** It has long been known that plane wave solutions of the cubic nonlinear Schrödinger Equation (NLS) are linearly unstable. This fact is widely known as modulation instability (MI), and sometimes referred to as Benjamin–Feir instability in the context of water waves. In 1978, I.E. Alber introduced a methodology to perform an analogous linear stability analysis around a sea state with a known power spectrum, instead of around a plane wave. This analysis applies to second moments, and yields a stability criterion for power spectra. Asymptotically, it predicts that sufficiently narrow and high-intensity spectra are unstable, while sufficiently broad and low-intensity spectra are stable, which is consistent with empirical observations. The bifurcation between unstable and stable behaviour has no counterpart in the classical MI (where all plane waves are unstable), and we call it Landau–Alber bifurcation because the stable regime has been shown to be a case of Landau damping. In this paper, we work with the realistic power spectra of ocean waves, and for the first time, we produce clear, direct evidence for an abrupt bifurcation as the spectrum becomes narrow/intense enough. A fundamental ingredient of this work was to look directly at the nonlinear evolution of small, localised inhomogeneities, and whether these can grow dramatically. Indeed, one of the issues affecting previous investigations of this bifurcation seem to have been that they mostly looked for the indirect evidence of instability, such as an increase in overall extreme events. It is also found that a sufficiently large computational domain is crucial for the bifurcation to manifest.

**Keywords:** nonlinear Schrödinger equation; modulation instability; Landau damping; Alber equation; rogue waves





## 1. Introduction

### 1.1. The Classical Modulation Instability and Rogue Waves

It is well known that plane wave solutions for focusing on the cubic nonlinear Schrödinger Equation (NLS) are linearly unstable, i.e., small perturbations will initially grow exponentially. This is called the Modulation Instability (MI), and was originally discovered in the context of optics [1] and water waves [2,3]. Today, there exists a vast literature on the various aspects of the problem. Mentioning only a few papers in a few directions, these include the nonlinear evolution beyond the initial exponential growth [4,5]; the implications for various physical problems where NLS appears as an approximate model [6,7]; analogous instabilities appearing in equations other than the NLS [5,8]; as well as a rigorous study of the well-posedness of the Cauchy problem for localised perturbation [9,10].

Beyond the initial exponential growth phase, it is now known that the perturbation often stops growing in amplitude and instead starts expanding its spatial extent [5,11]. This leads to the formation of a space–time cone with a robust pattern inside it, and the plane-wave background outside it. The simulations performed herein with various initial conditions (including low regularity ones) confirm the robustness of this structure, cf. Figure 1 and Appendix A.

The MI received renewed attention in the 2000s with the surge of interest in real-life oceanic rogue waves [12–20]. Since the NLS is widely used as an approximate model for unidirectional wave propagation [21], a nonlinear mechanism supporting the exponential

growth of localised perturbations may be relevant in rogue wave formation. It is therefore natural to explore this connection; the debate that ensued has been vigorous, with outlooks ranging from circumspect to polemical:

- "Under which conditions the Benjamin-Feir instability may spawn an extreme wave event: A fully nonlinear approach" [22]
- "Rogue waves in the ocean, the role of modulational instability, and abrupt changes of environmental conditions that can provoke non equilibrium wave dynamics" [23]
- "Rogue waves and their generating mechanisms in different physical contexts" [18]
- "Baseband modulation instability as the origin of rogue waves" [24]
- "Real world ocean rogue waves explained without the modulational instability" [25]

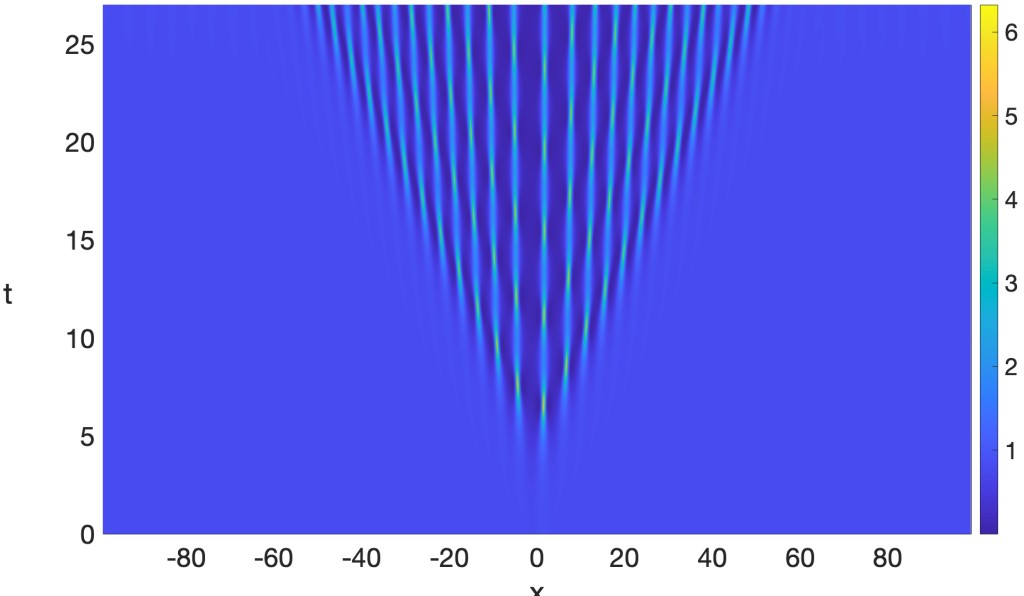

**Figure 1.** Position density of the wavefunction plotted over space and time. The initial condition is a plane wave with a localised perturbation, and it is evolved in time under the NLS. The NLS is solved numerically and periodic boundary conditions are used. More details about the setup and computation can be found in the Appendix A.

However, before one can earnestly engage with the question "is MI behind oceanic rogue waves?", there is a fundamental issue: realistic ocean waves are not exactly plane waves. One can still look for nonlinear effects *analogous* to the classical MI, but that would have to be a linear stability analysis around *a realistic sea state*. In other words, the messier nature of realistic sea states has to be taken into account somehow. This programme was launched in 1978 by I. E. Alber [26]. A brief overview, accessible to a general audience, can be found here [27].

### 1.2. The Alber Equation and the Landau–Alber Bifurcation

What does a typical sea state look like? The consensus among ocean engineers is that a typical sea state can be modelled as a typical realisation of a stationary and homogeneous random process with known autocorrelation (especially in deep water, away from topographical features that may drive the dynamics in localised ways) [28–31]. This is a powerful approach that provides a constructive way to estimate the probabilities of events, while at the same time being data-driven. Indeed, a vast amount of data available for ocean waves is in the form of second moments (autocorrelations, or equivalently their Fourier transforms, power spectra). The advantages of practical computations based on data are crucial and this methodology is the state of the art in ocean engineering and marine safety today [32].

Alber's idea in [26] was to take a stationary and homogeneous random process with a known power spectrum as the background solution, add to it a perturbation, and investigate whether the inhomogeneity grows under NLS dynamics. This analysis was wholly performed on the second moment level, and a Gaussian moment closure was used (the assumptions underlying it are identical to those used when generating realisations from a power spectrum, i.e., that the sea surface is a Gaussian, mean-zero, circularly symmetric random process, cf. [33], Appendix B).

This linear stability analysis leads to an instability condition that only depends on the background spectrum (and not on the initial inhomogeneity). In [26], it was shown that exponentially growing modes appear if the instability condition is satisfied. It took 40 years, utilising recent advances in Landau damping, to show that in fact, if the instability condition is not satisfied, the inhomogeneity disperses and does not grow [33]. In fact, the Landau-damping-type results show something more: *in the stable case, the nonlinearity plays a very small role in the evolution of the second moments*.

Thus, unlike the classical MI on a plane wave, for a background spectrum, both stability and instability are possible. This bifurcation between the *generalised MI* (genMI, i.e., the unstable regime of the Alber equation) and Landau damping (i.e., the stable regime) we describe as Landau–Alber bifurcation. It is interesting to ask what this analysis means for realistic ocean wave spectra. A study based on fitted JONSWAP spectra for the North Atlantic Scatter Diagram [33] found that, according to the linear stability condition, the sea states present for 99.8% of the time are expected to be in the stable regime, while the remaining 0.2% are expected to be unstable (of course, the linear stability analysis may not completely control the nonlinear dynamics, especially in borderline cases). This finding also goes a long way to explain why linear statistical methods for ocean waves are so successful, even in cases where the nonlinear terms are significant on the phase-resolved level, i.e., in the evolution of individual wave peaks and troughs. In fact, in the vast majority of cases, the nonlinearity has a very small impact on the *phase-averaged statistics* of sea states. That said, a probability of 0.2% of encountering an unstable sea state is not negligible from a marine safety perspective (at any given moment, there are tens of thousands of large vessels at sea).

One can very roughly describe the instability condition as follows: any regular bell-shaped spectrum will become unstable if it is sufficiently narrow or intense (in the sense that it is large enough and total integral); and conversely, it will become stable if it is sufficiently broad or low intensity. This is in very good agreement with the ocean engineering empirical understanding and Monte Carlo simulations [34–36]. In that context, it must be noted that realistic sea states with higher intensity also tend to have more narrow spectra, i.e., large storms are inherently much more likely to be unstable than milder sea states (also see the discussion on JONSWAP spectra in Section 3 and Figure 2).

However, an abrupt bifurcation between a stable and an unstable regime has not been found to date when looking at solutions of the NLS or other, more hydrodynamically accurate equations. When a systematic numerical study was undertaken in [37], only a gradual transition to higher extremes as spectra become narrower was found. A similar gradual increase in the likelihood of rogue waves was reported in [35]. In this work, we specifically look for the onset of genMI in problems where spectra cross over from stable to unstable. That is, *we do not look at extreme events in general but for genMI events in particular.* The precise definition of a genMI event is given in the next Section.

### 1.3. Onset of Generalised Modulation Instability on the Phase-Resolved Level

The main objective of this paper was to numerically investigate whether there is a sudden bifurcation between Landau damping and generalised modulation instability, and what that would look like on the phase-resolved level. A key novelty of this work is that we look for the onset of genMI directly: the analysis indicates that a localised perturbation will grow when genMI is present. We will therefore introduce a localised perturbation, and keep track of whether it grows or disperses. Thus, we define a "generalised modulation

instability event" as be the rapid growth of perturbation of a known, homogeneous solution. More precisely, consider a background solution of the NLS

$$iu_t + p\Delta u + q|u|^2 u = 0, \qquad u(x,0) = u_0(x) \tag{1}$$

where $u_0(x)$ is a typical realisation of a sea state with a known power spectrum. Then, one can consider adding a small, localised perturbation, $u_0(x) \mapsto u_0(x) + \delta_0(x)$. This would give rise to the initial value problem of finding $v(x,t)$ such that

$$iv_t + p\Delta vs. + q|v|^2 vs. = 0, \qquad v(x,0) = u_0(x) + \delta_0(x). \tag{2}$$

By taking the difference, we can recover the inhomogeneity $\delta(x,t)$,

$$\delta(x,t) := v(x,t) - u(x,t). \tag{3}$$

In this sense, we will say that we observe genMI if the localised perturbation $\delta(x,t)$ has an initial stage of clear exponential growth, and Landau damping if $\delta$ does not grow significantly and disperses. As we will see, there are cases exemplary of these two behaviours, as well as an intermediate regime for borderline spectra.

It must be noted that, if $u_0$ was a plane wave, $u_0 = A$, then $u(x,t) = Ae^{iqA^2t}$ and Equation (3) reduces to a particular way of looking at the classical modulation instability problem. In that case, the inhomogeneity $\delta$ would always initially grow. In fact, working this out can reveal some interesting insights, closely related to recent advances in the area [4,11]. We do this in Appendix A.

## 2. Main Results

Phase-resolved simulations are performed for various sea states generated by JONSWAP spectra. The scalings of JONSWAP spectra are discussed in Section 3, and the details of the computations are discussed in Section 4.

The idea is that a typical realisation of (the envelope for) a realistic sea state evolves over time, and then a localised perturbation is added to the same initial condition, as discussed in Equations (1)–(3). By taking the difference of an exact versus perturbed wavefunction, we can see what happened to the inhomogeneity.

The main result of this paper is that, for the spectra expected to be stable (according to Alber's stability condition), the inhomogeneity disperses, or only grows moderately for the cases closer to instability. Moreover, for the spectra expected to be unstable, the inhomogeneity grows rapidly and dominates the background solution. For the very stable and very unstable cases, we observe the exemplary behaviour of Landau damping and genMI, respectively. For the intermediate cases, it seems that nonlinear Landau damping does not hold (i.e., the linearised dynamics do not capture the fully nonlinear evolution of the sea state). However, the crucial thing is that, even though the inhomogeneity grows in some stable cases, it still remains smaller than the background solution. Thus, there is a kind of nonlinear stability here that is not nonlinear Landau damping, and that is not currently well understood.

Figure 2 presents the $\gamma - \alpha$ plane for JONSWAP spectra. The parameter $\gamma$ controls how peaked the spectrum is, and the parameter $\alpha$ controls the intensity. More details about JONSWAP spectra can be found in Section 3. The question of the separatrix between the stable and unstable region was the subject of several papers, with [38,39] proposing specific separatrices. In [33], the exact separatrix was computed by exactly resolving Alber's condition for the first time. Fitted sea states from the North Atlantic Scatter Diagram [40] are plotted as blue stars; the vast majority is on the stable region (below and to the left from the separatrix), while some rare extreme cases do cross over into the unstable region. The parameter values $(\gamma_j, \alpha_j)$ for $j = 1, \ldots, 7$ that were used in the simulations of this paper are plotted as red squares. Spectra encoded in $j = 1, 2, 3$ are in the stable region; spectra

encoded in $j = 5, 6, 7$ are in the unstable region; and the spectrum of $j = 4$ is virtually on the separatrix.

In Figure 3, we keep track of how large the inhomogeneity $\delta(t)$ becomes for each case $j = 1$ through $j = 7$. More precisely, the amplification factor is defined as $\max_{t \in [0,T]} \frac{\|\delta(t)\|_{L^p}}{\|\delta_0\|_{L^p}}$, and we take $p \in \{2, 4, \infty\}$. The final time $T = 8$ was used; and that seems to be sufficient for the inhomogeneity to reach its maximum size. The main finding is that the inhomogeneity stays $o(1)$ when the background spectrum is in the stable region $j = 1, 2, 3$, and becomes $O(1)$ when the spectrum is in the unstable region $j = 5, 6, 7$. In fact, in the unstable cases, $\delta$ reaches maximum values 2–3 times larger than the background solution at its largest. These maxima are localised, i.e., they can be thought of as rogue waves. More details about the setup and computation can be found in Section 4. These results are somewhat sensitive to the size of the computational domain and initial inhomogeneity, and hence a more extensive numerical simulation is warranted.

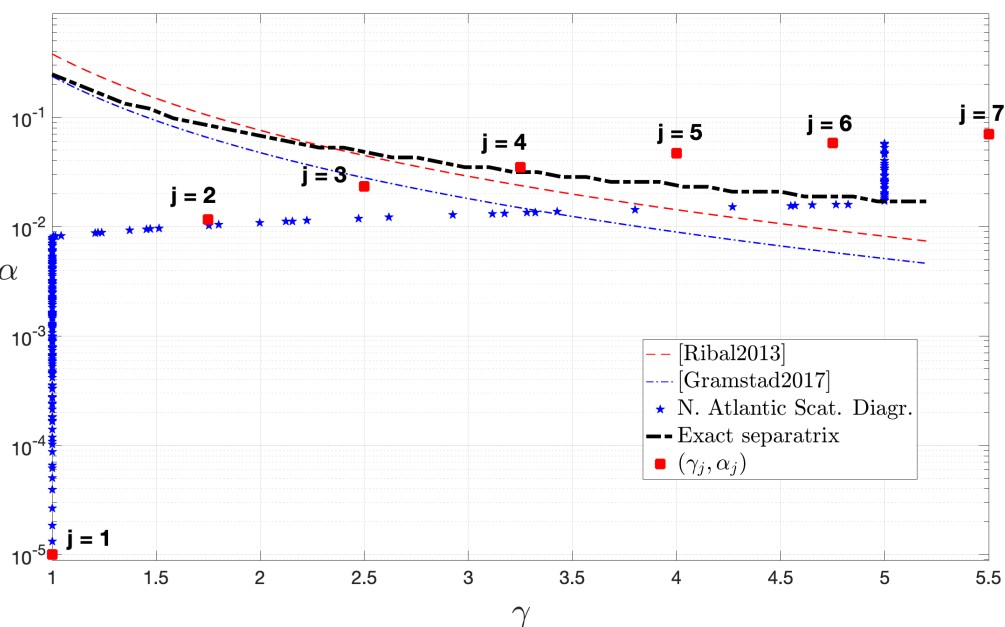

**Figure 2.** The $\gamma - \alpha$ plane for JONSWAP spectra. The exact separatrix between the stable and unstable region is shown and previously proposed separatrices are also included. The blue stars correspond to measured sea states from the North Atlantic Scatter Diagram. The red squares are the points used in the simulations for this paper $(\gamma_j, \alpha_j)$ for $j = 1, \ldots, 7$ [38,39].

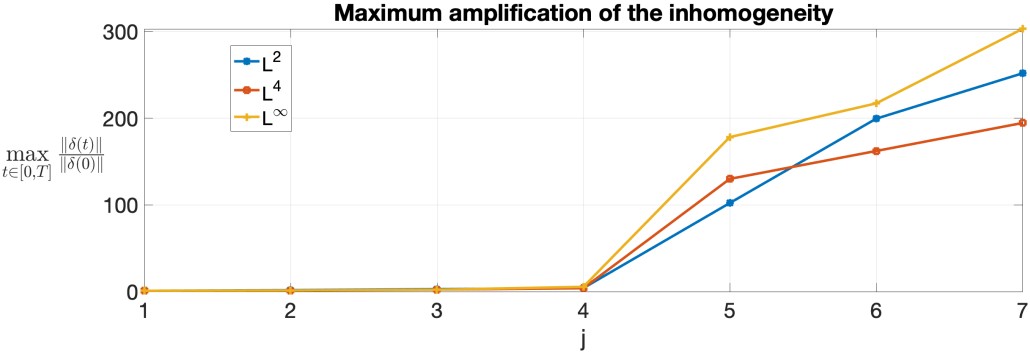

**Figure 3.** The maximum amplification factor for the inhomogeneity $\delta$ is plotted, in the $L^2$, $L^4$ and $L^\infty$ norms. $T = 8$ was used in all cases.

## 3. Scaling of JONSWAP Spectra and the $\gamma - \alpha$ Plane

JONSWAP spectra resolved over wavenumbers are typically written as

$$S(k) = C(k_0)\frac{\alpha}{2k^3}\mathrm{e}^{-\frac{5}{4}(k/k_0)^{-2}}\gamma^{\exp\left[-\left(1-\sqrt{k/k_0}\right)^2/2\sigma^2\right]}, \tag{4}$$

where $\sigma = \sigma(k) = 0.07$ if $k < k_0$ and $\sigma(k) = 0.09$ if $k \geqslant k_0$. The parameters have straightforward interpretations:

- $\alpha$ controls the power of the sea state, namely the significant wave height $H_s$, which is proportional to $\sqrt{m_0} = \sqrt{\int S(k)dk}$;
- $\gamma$ controls the peakedness of the spectrum, with larger values of $\gamma$ leading to more sharply peaked spectra;
- $k_0$ is the carrier wavenumber, i.e., $k_0 = 2\pi/\lambda_0$ where $\lambda_0$ is a central wavelength for the sea state.

It is well known that typically $C$ scales with $C = Dk_0^2$, and thus the rms steepness of the sea state depends only on $\gamma$ and $\alpha$, as does the Alber-stability of the spectrum [33,39]. This is because a wave height, e.g., of 6 m is "very nonlinear" if it has a wavelength of 60 m (and thus a steepness of ∼0.1), but the same wave height with a wavelength of 600 m is "very linear". Thus, it is typical to take out the $C(k_0)$ from the parameter $\alpha$ and work with non-dimensional wavenumbers $k' = k/k_0$, leading to a parametrisation of the problem with $\gamma$ and $\alpha$ only.

This now leads to each point of the $\gamma - \alpha$ plane being associated with a rms steepness for the sea state. This is the most direct physical quantity of "how nonlinear" the sea state is expected to be. Both the blue-and red-dashed lines in Figure 2 (the separatrices proposed by [38,39]) are contours of the steepness. We see that the actual separatrix is found to be broadly similar to a contour of the steepness too, although itself it is not a contour of the steepness.

In the end, both steepness and Alber stability do not depend on $k_0$ and can be exclusively thought of as a question of $\gamma$ and $\alpha$. We will see, however, some further subtle points on this in Section 4.2.

## 4. Numerical Results

### 4.1. The Numerical Scheme

We approximate the full continuous problem

$$iu_t + p\Delta u + q|u|^2u = 0, \qquad u(x,0) = u_0(x) \quad \forall x \in \mathbb{R} \tag{5}$$

with the periodic problem

$$\begin{aligned} iu_t + p\Delta u + q|u|^2u = 0, \qquad u(x,0) = u_0(x) \quad \forall x \in [-L, L], \\ u(-L,t) = u(L,t), \ \partial_x u(-L,t) = \partial_x u(L,t) \qquad \forall t > 0 \end{aligned} \tag{6}$$

where the initial data $u_0$ are also (made to be) compatible with the periodic BC. This is widely accepted common practice as long as $L$ is large enough. We proceed to discretise problem (6) as

$$\begin{aligned} i\frac{U^{n+1}-U^n}{\tau} + p\mathfrak{D}U^{n+\frac{1}{2}} + q\Phi^{n+\frac{1}{2}}U^{n+\frac{1}{2}} = 0, \\ \Phi^{n+\frac{1}{2}} = 2|U^n|^2 - \Phi^{n+\frac{1}{2}}, \end{aligned} \tag{7}$$

where

$$U^{n+\frac{1}{2}} := \frac{U^{n+1} + U^n}{2},$$

$\tau$ is the timestep and $U^n \approx u(t^n) = u(n\tau)$, $\Phi^{n+\frac{1}{2}} \approx |u(t^{n+\frac{1}{2}})|^2$. By updating the nonlinearity $\Phi^{n+\frac{1}{2}}$ in a decoupled way from the wavefunction $U^n$, we avoid having to use iterations,

i.e., the scheme is only linearly implicit. This is a second-order scheme, and moreover, it satisfies the energy and mass conservation on the discrete level [41].

To properly introduce the discrete Laplacian $\mathfrak{D}$, first consider a continuous periodic function on $[-L, L]$,

$$f : [-L, L] \to \mathbb{C}, \qquad f(-L) = f(L), \ \partial_x f(-L) = \partial_x f(L)$$

which is uniformly sampled,

$$f_m = f(x_m), \qquad x_m = -L + m\frac{2L}{M}, \qquad m = 0, 1, \ldots M - 1.$$

Then, $f_m$ can be used interchangeably with its discrete Fourier coefficients,

$$f_m = \sum_{l=-\frac{M}{2}}^{\frac{M}{2}} a_l e^{\frac{2\pi i}{2L} lm}, \qquad a_l = \sum_{j=0}^{M-1} f_j e^{\frac{2\pi i}{M} lj}.$$

The discrete Laplacian $\mathfrak{D}$ is defined as

$$\mathfrak{D} : f_m \mapsto \sum_{l=-\frac{M}{2}}^{\frac{M}{2}} \left(\frac{2\pi i l}{2L}\right)^2 a_l e^{-\frac{2\pi i}{2L} lm} = \sum_{j=0}^{M-1} \left[ \sum_{l=-\frac{M}{2}}^{\frac{M}{2}} \left(\frac{2\pi i l}{2L}\right)^2 e^{\frac{2\pi i}{M} lj} e^{-\frac{2\pi i}{2L} lm} \right] f_j.$$

This is similar to many existing trigonometric interpolation methods [42].

The computational domain used in the runs that follow is $x \in [-99, 99]$, and 4096 points are used in spatial discretisation. The timestep is $\tau = 0.005$ and the time of the computation is $t \in [0, 8]$ in all simulations reported in Figures 3–5.

*4.2. Implementation of the Initial Conditions and Periodisation*

As it is well known [21], the coefficients $p$ and $q$ in the NLS (1) for ocean waves are given by

$$p = \frac{\sqrt{g}}{8k_0^{\frac{3}{2}}}, \qquad q = \frac{\sqrt{g}}{2} k_0^{\frac{5}{2}}. \tag{8}$$

We will therefore need to select a wavenumber $k_0$ in order to perform a concrete numerical experiment.

We will proceed to generate a JONSWAP spectrum with carrier wavenumber $k_0$ and parameters $\alpha_j$ and $\gamma_j$, $j = \{1, \ldots, 7\}$ as in Equation (4). The choice of $(\gamma_j, \alpha_j)$ is visualised in Figure 2, in the context of the stable and unstable regions as well as of the measured data.

With the spectrum fixed, we will create a typical realisation of the sea surface elevation consistent with that spectrum. This is achieved by setting

$$\eta_0^{(j)}(x) = \sum_{n=1}^{N} A_n \exp(ik_n x) \quad \text{where} \quad A_n = Z_n \sqrt{2S_j(k_n)\Delta k_n}, \tag{9}$$

where $\Delta k_n = k_{n+1} - k_n$ is the grid spacing between the discrete wavenumbers and where $Z_n$ is an independent complex standard normal variable. In other words, the real and imaginary parts of $Z_n$ are normally independent distributed random variables with zero mean and variance $1/2$; equivalently, $|Z_n|$ are Rayleigh distributed with the parameter $\sigma = 1/\sqrt{2}$ so that $E[|Z_n|^2] = 1$ and $E[|A_n|^2] = 2S_j(k_n)\Delta k_n$, and the phases $Arg(Z_n)$ are uniformly distributed on $[0, 2\pi)$. This is the state-of-the-art method to produce realisations of sea states with a known power spectrum [35]. Finally, the envelope of the sea state is taken by modulating back the carrier wavenumber, $u_0^{(j)} = e^{-ik_0 x} \eta_0^{(j)}(x)$.

Moreover, to ensure that the initial condition $u_0^{(j)}$ is appropriate for simulation, as discussed in Section 4.1, we need to make sure that it has a period of $2L$, which is compatible with our finite computational domain. This is ensured by selecting $k_n \in \{\frac{m\pi}{L}, m \in \mathbb{N}\}$.

The constant $D$ appearing in the JONSWAP spectrum ($C(k_0) = k_0^2 D$ in Equation (4)) is uniformly fixed in $j$ so that the steepness values end up ranging between $0.8 \cdot 10^{-3}$ for $j = 1$ and $0.105$ for $j = 7$.

In this series of numerical experiments, the initial inhomogeneity is taken to be

$$\delta_0^{(j)}(x) = W^{(j)} \cdot 0.04 \cdot \mathrm{sinc}(x), \tag{10}$$

where $W^{(j)}$ is the rms amplitude of $u_0^{(j)}$, namely

$$W^{(j)} = \sqrt{\frac{\int_{x=-L}^{L} |u_0^{(j)}|^2 dx}{\int_{x=-L}^{L} dx}}. \tag{11}$$

*4.3. Discussion of the Results*

We keep track of the stability of the background sea state primarily through the growth (or lack thereof) of the inhomogeneity $\delta$ in time, as was discussed in Equations (1)–(3).

In Figure 4, we see various norms of inhomogeneity as functions of time for $j = 1$ (the most stable case) and $j = 7$ (the most unstable case). For $j = 1$, we see higher $L^p$ norms decreasing while the $L^2$ norm stays constant; this is evidence of dispersion, i.e., a simultaneous decrease in point values and spreading out of the function, i.e., an expansion of its support. These two effects completely balance out for the $L^2$ norm, as expected. That is, the inhomogeneity basically behaves as a solution of the free-space Schrödinger equation. On the other hand, for $j = 7$, we see an initial rapid exponential growth of the inhomogeneity in all norms, which then matures and plateaus. The mature inhomogeneity locally dominates the background solution (i.e., its $L^\infty$ norm is larger than that of the background solution). The slight upward drift of the $L^2$ norm after the initial exponential growth phase indicates some growth in the effective support of the inhomogeneity (observe that such a drift is absent in the $L^\infty$ norm). This is broadly analogous to the behaviour of the classical MI, cf. Figure 1.

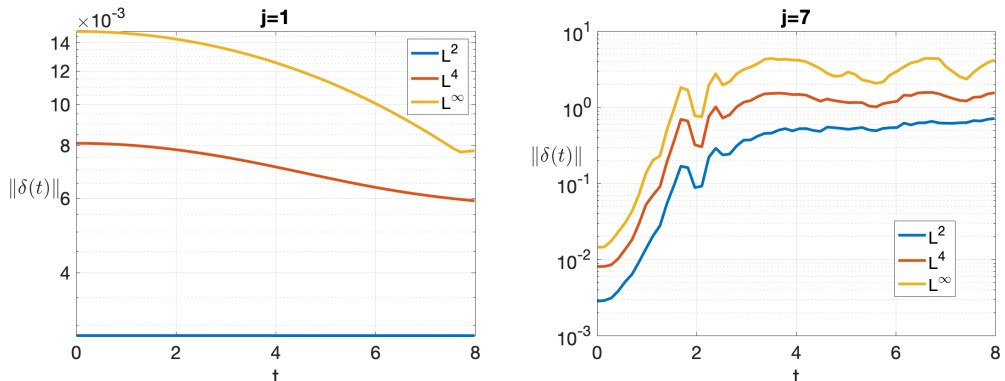

**Figure 4.** Size of the inhomogeneity as a function of time, for $j = 1$ and $j = 7$. The $L^2$, $L^4$, and $L^\infty$ norms are used.

In Figure 5, we see the change in behaviour as the background spectrum crosses over, from the stable region to the unstable one. For $j = 3$, the stable case that is closest to the separatrix, we see the inhomogeneity growing a little without dispersing, i.e., nonlinear Landau damping does not seem to hold. Nonetheless, the inhomogeneity stays at $o(1)$. We should recall that the "stable region" is characterised by a linear stability analysis. This absence of *nonlinear Landau damping* as we move closer to the separatrix is perhaps expected. After all, the nonlinear evolution stays close to the linearised dynamics only

under strong smallness assumptions for the inhomogeneity. If one extrapolates nonlinear Landau damping results [43] and asks how such a result for the Alber equation might look like, a smallness condition of the form

$$\frac{\|\delta_0\|_{\Sigma^r}}{\kappa^2} \leqslant \epsilon \tag{12}$$

is expected for the initial inhomogeneity, where $\Sigma^r$ would be a norm of the form

$$\|\delta\|_{\Sigma^r} = \sum_{a+b \leqslant r} \|x^a \partial_x^b \delta\|_{L^2}, \tag{13}$$

$\epsilon$ a small constant and $\kappa$ a constant that scales with the distance of the spectrum from being unstable. Thus, for spectra that are somewhat close to the separatrix, this might end up being an extremely stringent constraint that is unrealistic. A striking feature here is the level at which the inhomogeneity plateaus, which is still $o(1)$.

Continuing in Figure 5, for $j = 4$, a spectrum virtually on the separatrix, we see some irregular growth and the inhomogeneity remains smaller than the background solution. The first unstable case is for $j = 5$; we see a slow exponential growth and the inhomogeneity does eventually become $O(1)$. For stronger instability in the case of $j = 6$, we see a clear separation of the initial, exponential growth phase and the mature phase afterwards—in fact, it looks quite similar to the case $j = 7$.

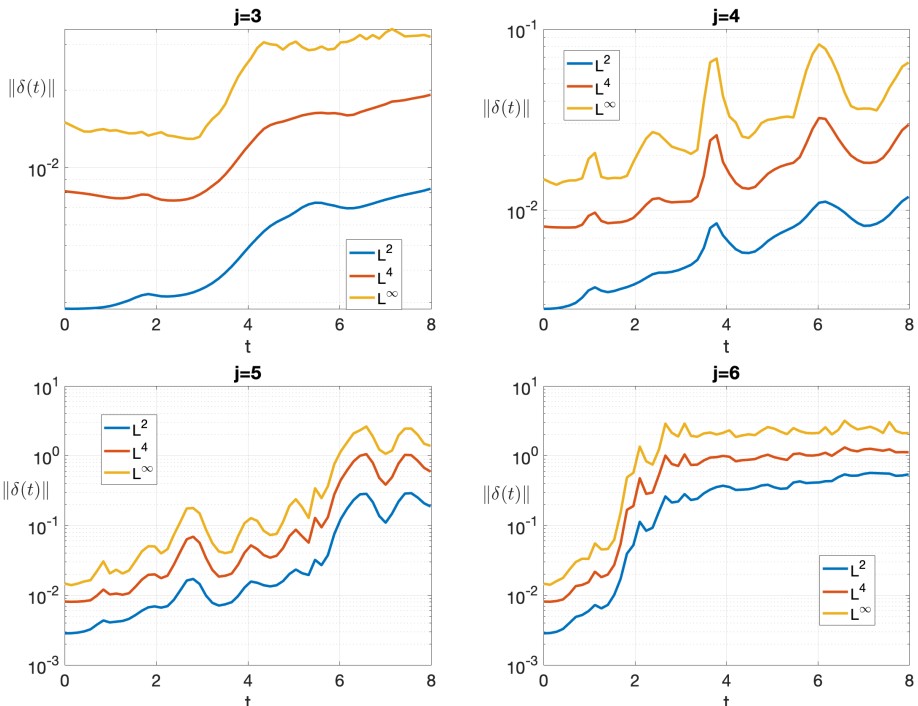

**Figure 5.** Size of the inhomogeneity as a function of time, for $j = 3, 4, 5, 6$. The $L^2$, $L^4$, and $L^\infty$ norms are used.

### 4.4. Dependence on the Computational Domain

Many choices are required in order to generate a path function consistent with a given sea state (i.e., compatible with a given power spectrum) and investigate its Alber stability as in Equations (1)–(3). These were discussed in Section 4.2.

It must be pointed out that $k_0$ does not affect the Alber stability analysis for JONSWAP sea states [33,39], as its total effect (in the spectrum and the coefficients of the NLS (1)) cancels out. Hence, the choice of wavenumber $k_0$ may be thought of as inconsequential.

However, since we worked with a fixed computational domain of $x \in [-99, 99]$, the significance of the wavenumber is that it controls how many wavelengths are there in

our computational domain. This turns out to be a major decision—and one should recall that the theory applies to the infinite line, $x \in (-\infty, \infty)$. It is common practice to use a "large" computational domain with periodic boundary conditions, and usually this step does not invite very much scrutiny. However, here we find that the question "how large is large enough" is crucial. For these results, we have ~126 wavelengths $\lambda_0 = 2\pi/k_0$ in the computational domain. This breaks down to ~32 points per wavelength which, for a Fourier-spectral spatial discretisation method, is considered quite well resolved. The inhomogeneity also has an effective support comparable to the wavelength $\lambda_0$.

When there are much fewer wavelengths in the computational domain, the inhomogeneity always disperses and we do not see any clear bifurcation. Furthermore, the initial inhomogeneity seems to play a role in the appearance of the instability. It might be the case that the slow decay of the *sinc* function actually plays a role in activating the instability.

### 5. Conclusions and Further Work

We presented a methodology for the detection of generalised modulation instability in sea states generated by a known power spectrum, using phase-resolved simulation. Using that, we recovered direct evidence for an abrupt Landau–Alber bifurcation, i.e., the abrupt onset of generalised MI once the power spectrum becomes sufficiently narrow and intense.

To the best of the author's knowledge, this abrupt bifurcation has not been observed before. Part of the explanation is that, *here, we look in a focused way for genMI events, not merely for extreme events of any origin*. Indeed, there may be extreme events due to many mechanisms (linear and nonlinear), and there is evidence that such extreme events gradually increase as the spectra become less stable [35,37]. Here, we find that unstable sea states are expected to have genMI events that will produce extreme events and most likely rogue waves. We do not claim that all rogue waves are due to genMI.

More subtle reasons for this bifurcation not being observed before may have to do with the size of the computational domain used and the kind of inhomogeneity used. It might be interesting to consider using other boundary conditions on large enough domains, since *periodisation is known to have a stabilising effect; after all, there is no MI for the NLS on the torus*.

A next natural step is to perform more extensive simulations (larger computational domains, more realisations of the initial conditions). Another direction is to extend this analysis to two-dimensional problems, including crossing seas (i.e., energy arriving from different directions). More broadly, one could look for numerical methods that would allow this kind of simulation to be made without periodising the problem (e.g., using some kind of near/far field decomposition). Periodisation is not part of the theory nor of the real-life problem, and it might be artificially suppressing nonlinear and unstable features of the problem.

**Funding:** This research received no external funding.

**Data Availability Statement:** The computed wavefunction $u(x,t)$ and inhomogeneity $\delta(x,t)$ can be found in the author's github repository, https://github.com/aathanas/LandauAlberData (accessed on 21 December 2022).

**Acknowledgments:** The author would like to thank O. Gramstad (DNV), A. Babanin (University of Melbourne), and K. Spyrou (NTUA) for helpful discussions and their encouragement while working on this problem.

**Conflicts of Interest:** The author declares no conflict of interest.

### Appendix A. Simulation of the Classical MI

In [5,11], the nonlinear evolution of MI is examined. It was found that, once the localised inhomogeneity grows enough in amplitude, it stops growing and it spreads out its spatial support. That way, a space–time cone is formed, with an homogeneous pattern of fixed amplitude inside the cone, and approximately zero outside. Here, we reproduce

this behaviour for various initial inhomogeneities added to the plane wave solutions of the cubic NLS. Some numerical issues related to periodisation are highlighted in the process.

More specifically, we consider the IVP

$$i\partial_t v_j + p\Delta v_j + q|v_j|^2 v_j = 0, \qquad v_j(x,0) = A + \delta_j^0(x) \tag{A1}$$

and extract the inhomogeneity

$$\delta_j(x,t) = v_j(x,t) - A e^{iqA^2t}. \tag{A2}$$

We use the parameters

$$p = \frac{1}{2}, \qquad q = 1, \qquad A = 0.99, \qquad t \in [0,35]. \tag{A3}$$

A computational domain of $[-99, 99]$ was used with 4096 points in the spatial discretisation and periodic boundary conditions. The timestep used was $\tau = 0.007$. The four initial inhomogeneities used are

$$\delta_1(x) = 0.05\, e^{-x^2} + 0.01\, \text{sech}(x), \tag{A4}$$

$$\delta_2(x) = 0.05\, e^{-x^2} + 0.01\, i\, e^{-(x+0.021)^2}, \tag{A5}$$

$$\delta_3(x) = 0.05\, e^{-x^2} \mathfrak{N}(x) + 0.01\, i\, e^{-(x+0.021)^2}, \tag{A6}$$

$$\delta_4(x) = 0.05\, e^{-x^4} x^3 + 0.01\, i\, e^{-(x+0.021)^2}, \tag{A7}$$

where $\mathfrak{N}(x)$ is discrete white noise (for each discretisation point $x_m$, a normal random variable $X_m \sim \mathcal{N}(0,1)$ multiplies the smooth envelope). The position density $|v_j(x,t)|^2$ is plotted as a function of $x, t$ in Figure A1. We observe that, qualitatively, all smooth initial conditions lead to very similar patterns. Even the non-smooth $\delta_3$ essentially exhibits the same kind of pattern: the space–time cone of expanding inhomogeneity. A somewhat surprising feature is that, around time $t = 30$, in all cases, the cone seems to suddenly expand and cover all the computational domain. The solutions seem to not change if we apply moderate refinements in space and time. This is likely an artefact of periodisation, and highlights the need to investigate the role of periodisation when trying to simulate problems on the real line with large computational domains.

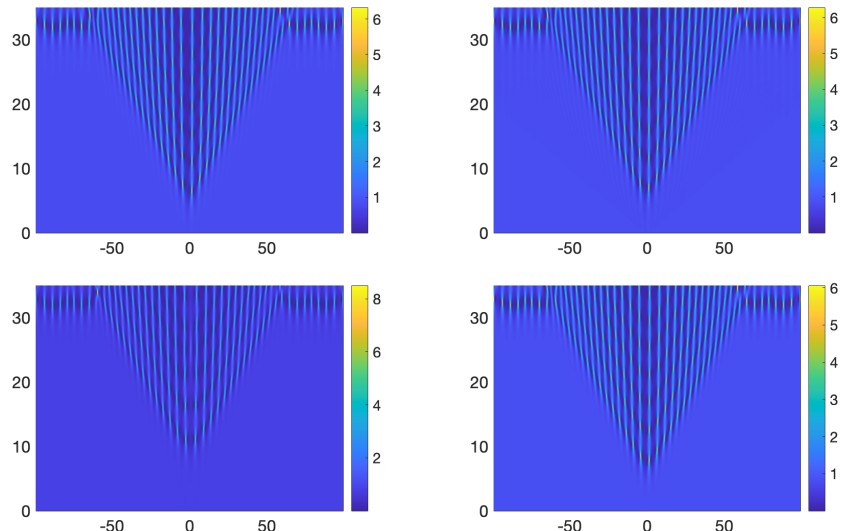

**Figure A1.** In all plots, the horizontal axis is space, vertical axis is time. **Top left**: $|v_1(x,t)|^2$. **Top right**: $|v_2(x,t)|^2$. **Bottom left**: $|v_3(x,t)|^2$. **Bottom right**: $|v_4(x,t)|^2$. In all cases $x \in [-99, 99]$, $t \in [0,35]$. The solutions conserve mass and energy in time.

It is very interesting to note that in [5], MI is found for other kinds of dispersive equations, in addition to the NLS. One could investigate whether equations, such as, e.g., [44–47] also exhibit MI.

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
