# Peer review of "Phase Resolved Simulation of the Landau–Alber Stability Bifurcation"

_fluids, doi:10.3390/fluids8010013_

Round 1
Reviewer 1 Report
The authors presented an interesting numerical simulation on Phase resolved simulation of the Landau-Alber bifurcation.
What is the novelty of the current study? To be clearly stated.
What are the actual applications of the considered configuration?
More details on the numerical method are to be provide.
The results are to be described with more details.
The discussion is to be improved by adding Physical interpretations.
Author Response
I would like to thank the reviewer for the helpful comments and suggestions. In particular, extensive changes have been made to the paper in order to make sure that I examine a problem that is closely tied to a physical application. It is the same kind of analysis and results, but now the application to a real-life problem is direct. The abstract has been modified appropriately to describe the new version of the paper.
Detailed point by point responses follow:
On the novelty of this work: This is the first time that the bifurcation predicted by Alber is captured on the phase-resolved level. This is now much more clearly discussed in Section 1.2 of the revised paper, along with earlier investigations and why these did not capture the bifurcation. The main results are also summarised in Section 2.
Applications and physical interpretation: in order to help connect the results with their natural physical context, a brief discussion presenting the links between modulation instability and rogue waves has been added in Section 1.1. Oceanic wave extreme events is the natural context for this problem, and now this is much more clearly emphasised. Furthermore, instead of the somewhat artificial gaussian spectra, realistic JONSWAP spectra of ocean waves are used. These are introduced in Section 3, and the main results summarised in Section 2 with sufficient context about the JONSWAP spectra and what is known about their stability. The implications for real life ocean waves are also discussed in the conclusion section.
Details on the computation: The numerical method is moved from the Appendix to Section 4.1. Also, a detailed discussion of the generation of the initial conditions is added in Section 4.2. Some subtle issues related to the computation are further discussed in Section 4.3.
More detailed discussion of the results: Detailed discussion of the results is added in Section 4.3.
Reviewer 2 Report
The author explores direct evidence for a sudden bifurcation when the power spectrum of a homogeneous solution of the NLS equation becomes narrow enough. Such studies and numerical simulations are interesting and important in applications of the considered NLS equation. Recently, various nonlocal integrable NLS equations have been proposed (see, e.g., Int J Appl Comput Math, 8(2022), no.4, 206 and Proc Ameri Math Soc, Ser B, 9(2022), 1-11) and their soliton solutions could be formulated by the Riemann-Hilbert technique. It should be interesting to remark about MI problems for such nonlocal NLS equations. The manuscript successfully analyzes a bifurcation phenomenon of the Cauchy problem of the considered nonlinear dispersive wave model with specific initial values. I would, therefore, like to recommend publishing a slightly amended manuscript in the journal.
Author Response
I would like to thank the reviewer for his careful reading and helpful comments. The question of exploring the appearance of modulation instability in further equations, such as those proposed by the reviewer and some more, has been added in Appendix A.
Also, some broader changes have been made to the paper, to make sure that the cases studied in it are closely related to physical applications. (This was mainly motivated by thoughtful questions by the other reviewer.)